# Doping β-TCP as a Strategy for Enhancing the Regenerative Potential of Composite β-TCP—Alkali-Free Bioactive Glass Bone Grafts. Experimental Study in Rats

**DOI:** 10.3390/ma12010004

**Published:** 2018-12-20

**Authors:** Manuel M. Ferreira, Ana F. Brito, Daniela Brazete, Inês C. Pereira, Eunice Carrilho, Ana M. Abrantes, Ana S. Pires, Maria J. Aguiar, Lina Carvalho, Maria F. Botelho, José M.F. Ferreira

**Affiliations:** 1Coimbra Institute for Clinical and Biomedical Research (iCBR) area of Environment Genetics and Oncobiology (CIMAGO), Faculty of Medicine, University of Coimbra, 3000-548 Coimbra, Portugal; m.mferreira@netcabo.pt (M.M.F.); ana.brito@reg4life.com (A.F.B.); eunicecarrilho@gmail.com (E.C.); mabrantes@fmed.uc.pt (A.M.A.); a.salome.pires@gmail.com (A.S.P.); mfbotelho@fmed.uc.pt (M.F.B.); 2Faculty of Medicine, University of Coimbra, 3000-548 Coimbra, Portugal; lcarvalho@huc.min-saude.pt; 3OssMed—Regeneration Technology, Biocant - Ass de Transf. de Tecnologia, Pq Tecnológico de Cantanhede, Núcleo 04, Lote 3, 3060-197 Cantanhede, Portugal; contact@ossmed.eu; 4Biophysics and Biomathematics Institute, IBILI-Faculty of Medicine of University of Coimbra, Coimbra 3000-548, Portugal; 5Department of Materials and Ceramic Engineering, CICECO, University of Aveiro, Campus Santiago, Aveiro, 3810-193 Aveiro, Portugal; d.s.b@live.ua.pt (D.B.); icarolina@live.ua.pt (I.C.P.); jmf@ua.pt (J.M.F.F.); 6Institute of Integrated Clinical Practice, Faculty of Medicine, University of Coimbra, 3000-548 Coimbra, Portugal; 7Institute of Pathological Anatomy, Faculty of Medicine, University of Coimbra, 3000-548 Coimbra, Portugal; iap@fmed.uc.pt

**Keywords:** composite bone grafts, calcium phosphates, implants, dentistry, biomedical engineering, osteogenesis

## Abstract

The present work aims at evaluating the potential gains derived from partially replacing calcium in resorbable β-tricalcium phosphate (β-TCP) by two different molar percentages of strontium (5, 10) and zinc (1, 2), concomitantly with a fixed molar percentage (0.5) of manganese. Synthetic granular composite bone filling grafts consisting of doped β-TCP and an alkali-free bioactive glass were prepared and implanted in ~4 mm diameter bone defects drilled in the calvaria of Wistar rats used as animal models. The animals were sacrificed after 9 weeks of implantation and the calvaria was excised. Non-manipulated bone was used as positive control, while empty defects were used as a negative control group. The von Kossa staining revealed an enhanced new bone formation with increasing doping levels, supporting the therapeutic effects exerted by the doping elements. The percentage of newly formed bone was similar when the defects were filled with autologous bone, BG (previous results) or 3TCP2/7BG, which indicates that the latter two are excellent candidates for replacement of autologous bone as bone regeneration material. This finding confirms that doping with suitable doses of therapeutic ions is a good strategy towards transposing the bone graft materials to biomedical applications in humans.

## 1. Introduction

Problems associated with the skeletal system affect millions of people around the world, especially in people over fifty years old. Fractures related to osteoporosis, severe injuries, different types of diseases and bone disabilities are among the various causes of chronic pain and bone loss, leading to morbidity and mortality of hundred millions of people [1]. Several types of bone graft materials have been used to restore damaged bones, including different types of calcium phosphates (CaPs) and biodegradable materials [2,3] in a variety of forms such as powders [4], scaffolds [5] and cements [6]. The similar chemical composition of CaPs to the inorganic part of hard tissues is considered as an advantage that justifies their selection for applications in orthopedics [7], dentistry [8,9], repair of fractures, cranium-maxillofacial reconstruction, spinal surgery, and ophthalmology [10].

An ideal bone substitute must be biocompatible, osteconductive to enable osseointegration, and osteoinductive to stimulate the mesenchymal stem cells to differentiate in bone-forming cells [11,12], Additionally, they must also undergo the degradation and resorption processes *in vivo* [13,14]. These relevant characteristics have been identified in several studies dealing with CaPs such as β-tricalcium phosphate (Ca_3_(PO_4_)_2_, β-TCP) [15,16,17,18,19,20,21] and hydroxyapatite (Ca_10_(PO_4_)_6_(OH)_2_, HA) [15,16,17,19]. The biodegradation mechanisms and resorption behavior *in vivo* of CaPs materials intended for bone repair and tissue engineering applications were recently reviewed [3,13]. Similarly, a number of other recent literature reports provide interesting evidence about the *in vitro* and *in vivo* performance of alkali-free bioactive glasses (BGs) [22,23,24,25,26,27].

The mineral part of the bone consists of inorganic salts and crystals [28]. Synthetic HA and β-TCP are prone to incorporate some trace doping elements into the crystalline lattices. Some trace elements existing in bone composition play essential roles in bone formation, growth and repair. Strontium (Sr), zinc (Zn) and manganese (Mn) are some of the most beneficial doping elements [28,29] commonly used to improve the physicochemical properties of the biomaterials [30]. The replacement of Ca^2+^ by Sr^2+^ in the crystalline structures of incorporation of HA and β-TCP enhanced the solubility of CaPs [31] stimulated osseointegration [32]. Preventing osteoporosis by reducing reabsorption by osteoclasts and consequently increasing bone formation and the mechanical resistance of the hard tissues, thus reducing the risk of bone fractures are other important benefits commonly reported [21,33,34]. Zinc plays an important role in bone metabolism [32], stimulating bone formation and regeneration, while improving bone mineralization [35,36,37,38] and cell adhesion [39]. Other studies showed that Zn-doping endows the bioactive bone grafts with anti-inflammatory activity [40]. This is of paramount importance to inhibit bacterial growth and biofilm formation and, therefore, for preventing infections at the surgical site, thus stimulating wound healing [41,42]. Manganese plays an important role as cofactor for several enzymes that are involved in extracellular matrix remodeling. It is also of great importance with regard to the binding affinity of integrins, receptors mediating the cellular interactions and promoting cell adhesion [43,44,45]. Previous *in vitro* studies have shown that low levels of Mn-doping in β-TCP powders exerted several stimuli towards a pre-osteoblastic cell line viability, proliferation and differentiation [46].

The ability of BGs to establish direct and strong chemical bonds with bone and soft tissues was firstly discovered by Larry Hench and co-workers for the well-known 45S5 Bioglass^®^ composition [47,48]. However, the high dissolution and degradation rates of 45S5 Bioglass^®^, mostly derived from its high alkali content (>24 mol%) [49,50] are often pointed out as some drawbacks. Moreover, 45S5 Bioglass^®^ hardly can be used for the fabrication of porous structures used as scaffolds for applications in bone regeneration and tissue engineering. As a matter of fact, the readiness to crystalize due to the extremely poor sintering ability, brings serious drawbacks in terms of mechanical properties [51,52,53,54]. These limitations can be overcome by using alkali free BGs [23,24,25,27,55,56] that possess several advantages listed in the Table 1. An alkali-free BG composition designed in the ternary system and consisting of (wt.%): 70 diopside (Di-CaMgSi_2_O_6_)–20 tricalcium phosphate (TCP-3CaOP_2_O_5_)–10 fluorapatite (FA-Ca_5_(PO_4_)_3_F) was selected. This composition was shown to exhibit a fast bio-mineralization rate, with the formation of a carbonated hydroxyapatite (HCA) surface layer after 1 h of immersion in simulated body fluid (SBF) being detected by XRD and FTIR analyses [23]. Since the bone bonding to living tissues is mediated by this HCA layer, this bioactive glass composition was suggestively designated as FastOs^®^BG to give a due account of this particular feature.

It was also demonstrated that FastOs^®^BG could induce and stimulate the differentiation of human mesenchymal stem cells (hMSCs) into bone-forming cells, even under the less favorable conditions (absence of osteogenic medium) [27]. When FastOs^®^BG was implanted in sheep as animal model it also revealed an excellent *in vivo* accomplishment [24] in comparison to 45S5 Bioglass^®^ used as control. In another recent study [22], the *in vivo* performances of FastOs^®^BG used as single material, or mixed with pure β-TCP were compared with each other, and with a commercial granular product, (adbone^®^BCP) supplied by Medbone^®^, Sintra Portugal. This last bone graft material is a biphasic calcium phosphate allegedly containing ~75% of HA and ~25% of β-TCP. The bone graft materials implanted in calvaria Wistar rats enabled to conclude that FastOs^®^BG alone gave the most interesting biological response among all the tested materials.

The aptitude of calcium phosphates as bone graft materials is very well-known and extensively documented [6,7,14,18,28,29,36,43,57,60]. Considering the potential benefits of doping elements in terms of the *in vitro* and *in vivo* performances, the present work aims at replacing the pure β-TCP in the composite mixtures by doped β-TCP powders and investigating if doping could bring further advantages in terms of bone regeneration in comparison to the non-doped counterpart. Powders of β-TCP with two molar concentrations of different doping elements: Sr (5, 10), Zn (1, 2), and a fixed amount of Mn (0.5) were prepared. Non-manipulated bone and the composite containing pure β-TCP were used as control materials.

## 2. Materials and Methods

### 2.1. Preparation of the Starting Metal-Doped β-TCP and FastOs^®^BG Powders

The β-TCP powders non-doped (TCP0) and doped with Sr, Zn and Mn ions were synthetized via wet chemical precipitation. The concentrations of the precursors were chosen in order to obtain a total (Ca + Sr + Zn + Mn)/P molar ratio of 1.5. The different dopant concentrations selected were as follows: β-TCP1 (5, 1, and 0.5 mol% of Sr, Zn and Mn, respectively) (TCP1); and β-TCP2 (10, 2 and 0.5 mol% of Sr, Zn and Mn, respectively) (TCP2). For this purpose, analytical grade precursor reagents were combined in the required molar percentages. The following precursors were used: calcium nitrate tetrahydrate (Ca(NO_3_)_2_ 4H_2_O, Quality Chemicals, Barcelona, Spain) for Ca, diammonium hydrogen phosphate ((NH_4_)_2_HPO_4_, Quality Chemicals, Barcelona, Spain) for P, strontium nitrate (Sr(NO_3_)_2_, Sigma-Aldrich, Darmstadt Germany) for Sr, zinc nitrate hexahydrate (Zn(NO_3_)_2_ 6H_2_O, Sigma-Aldrich, Darmstadt, Germany) for Zn, and manganese(II) nitrate hydrate (Mn(NO_3_)_2_xH_2_O, Sigma-Aldrich, Darmstadt, Germany) for Mn. The stock solutions containing the cationic and anionic species were prepared separately. After total dissolution of the precursors, the phosphate solution was drop wisely added to the cationic solution. The pH of the mixture was maintained at ~7 by carefully adding the required amounts of concentrated ammonium hydroxide (NH_4_OH) solution. The reaction was carried out for 3 h under controlled temperature and stirring conditions (30 °C, 1000 rpm).

The obtained solid precipitate was then separated from the supernatant solution through vacuum filtration and dried overnight in an oven at 100 °C. The dried powder was deagglomerated using a mortar and pestle and then calcined at 800 °C for 2 h. After calcination, the powder was deagglomerated by ball milling as reported elsewhere [60].

The FastOs^®^BG composition (38.49 SiO_2_-5.61 P_2_O_5_-36.07 CaO-19.24 MgO-0.59 CaF_2_, in mol%), was prepared by the melt-quenching route [20]. The batches (~100 g) were prepared by combining appropriate amounts of high purity silica (purity >99.5%), and precursor reagents for Ca-CaCO_3_ (>99.5%); Mg-MgCO_3_ (BDH Chemicals Ltd., London, UK, purity >99.0%); P-NH_4_H_2_PO_4_ (Sigma-Aldrich, Darmstadt, Germany, >99.0%), and F-CaF_2_ (Sigma Aldrich, Darmstadt, Germany, 325 mesh, >99.9%). The batches were thoroughly mixed and homogenized by ball milling during 30 min and then decarbonated by heating at a rate of 5 °C min^−1^ up to 900 °C, followed by 1 h dwell time at this temperature. The melting of the batches was performed in a Pt-10Rh crucible at 1570–1590 °C for 1 h using the same heating rate of 5 °C min^−1^. The molten glass was then quenched by pouring it in cold water to obtain the glass frit. Afterwards, the frit was dried in an oven at 100 °C and then milled in a high-speed agate mill for the time required to obtain a powder with a mean particle size ≤10 µm. The milled FastOs^®^BG frit revealed to be completely amorphous. The crystalline phases of non-doped and (Sr, Zn, Mn)-doped β-TCP powders were determined by X-ray powder diffraction (XRD) analysis (X’Pert PRO PANalytical, Almelo, Netherlands) and Fourier Transform Infrared Spectrometry (FT-IR) model Bruker Tensor 27 FT-IR; Bruker, Billerica, MA, United States).

### 2.2. Preparation of Porous Granular Composite Bone Grafts

Porous granular composites consisting of 3/7 volume ratio between the different calcium phosphate components (β-TCPn, n = 0, 1, 2), and FastOs^®^BG were prepared according to the procedure reported elsewhere [22]. Table 2 presents the detailed description of the individual components, the composite materials prepared therefrom, and their respective sample codes. These short names will be hereafter preferably used throughout the manuscript.

Suspensions containing a 3/7 volume ratio of (β-TCPn, n = 0, 1, 2)/FastOs^®^BG and a total solid loading of 60 vol.% were prepared by adding 0.4 wt.% of Targon 1128 as dispersing agent. TRECOMEX AET1 (Starkelsen Lyckeby AB, Blaklycke Jämjö Sweden—esterified potato starch with an average grain size of about 55 µm) was added as pore former agent in the same volume fraction as inorganic solids. The as-obtained suspensions were then mixed with a 3 wt.% sodium alginate solution (Sigma-Aldrich, Darmstadt, Germany) at the weight ratio of 0.8. The as-prepared mixtures were kept under stirring until reach complete homogenization. Afterwards, spherical granules were obtained by dropping the homogenized suspension into the setting CaCl_2_ solution. A spraying system consisting of a peristaltic pump (505S; Watson Marlow, Falmouth Cornwal, England) adjusted for a suspension flow rate of 9 rpm through a 1.5 mm nozzle diameter under a constant air pressure of 0.8 bar was used. The granules generated by the gradual gelation of sodium alginate with the Ca^2+^ ions were maintained in CaCl_2_ solution for about 10 min to allow their complete consolidation. The as-obtained granules were then separated from the solution by sieving, dried overnight at 80 °C, and then heat-treated to burnout the organics, and sintering at 800 °C. Considering that burnout of the organics is a slow process, a heating rate of 1 °C min^−1^ was used within the corresponding temperature range of 250–500 °C. The first and the last heat treatment steps up to 250 °C, and from 500–800 °C, respectively, were both conducted at the heating rate of 5 °C min^−1^ up to 800 °C. After a dwell time of 2 h at 800 °C, the furnace has been turned off, followed by natural cooling to RT as reported elsewhere [22]. The obtained granules were finally sterilized by autoclaving and safely packed before implantation.

### 2.3. XRD, FT-IR and Particle Size Distribution Analyses

A qualitative phase analysis of the inorganic powders was performed by X-ray diffraction. For this, a high-resolution X-ray diffractometer (X’Pert PRO PANalytical, Almelo, Netherlands)) equipped with Ni-filtered CuKα radiation (λ = 1.54056 Å) was used to collect the X-ray diffraction data within the 2θ range of 5–110° using a step size 0.02° and 96 s of counting time for each step. The ICDD cards numbers # 04 006 9376 for β-TCP and # 04 009 3876 for β-CCP were used as models for identifying the crystalline phases.

Infrared spectra were obtained by FT-IR (model Bruker Tensor 27 FT-IR; Bruker, Billerica, MA, USA). Each starting powder was mixed with KBr in the proportion of 1/150 (by weight) for 15 min and pressed into a pellet. Each infrared spectrum was the average of 128 scans collected at 4 cm^−1^ resolution at room temperature (RT).

Particle size distribution (PSD) and the average particle size (PS) of the powder were assessed using a laser diffraction particle size analyzer (COULTER LS230, Northewell Drive, Luton, England, Fullerton CA—Fraunhofer optical model).

### 2.4. Surgical Procedure

The regenerative potential of the different granules prepared was evaluated by animal experiments, using Wistar rats as an animal model (Figure 1). This study was performed according to the Declaration of Helsinki and in accordance with guidelines of the Portuguese Society of Animal Science Laboratory and of the Council for International Organization of Medical Sciences Ethical Code for Animal Experimentation. The study was approved by the Ethical Committee of the Faculty of Medicine of University of Coimbra, Coimbra, Portugal (protocol number: 005-CE-2014). In this study, ten thirteen-week old Wistar rats were used.

On the day of surgery (day 0) the rats were anesthetized by intraperitoneal administration of chlorpromazine (Largactil^®^, Laboratórios Vitória, Amadora 2700-326, Portugal) plus ketamine (Ketalar®, Pfizer, 2740-244 Porto Salvo, Portugal) 1:3 (0.1 mL/20 g) solution. A trichotomy of the skull cap was then performed, and the region was disinfected with povidone-iodine. A longitudinal incision with 3 cm was made with a scalpel # 12 to expose the calvaria bone. Then, on each side of the cranium, two bone defects of approximately 4.4 mm diameter were performed using a trephine bur with 4 mm of diameter and a motor with 5000 rpm under constant irrigation with saline solution (Solução isotónica de Cloreto de Sódio, Paracélsia, Indústria Farmacêutica, S.A. Porto, Portugal).

Five experimental groups were formed. The non-manipulated bone was used as positive control. The bone defect not filled (empty) was used as the negative control. In the other 3 experimental groups, the bone defects were filled with: 3β-TCP0/7FastOs^®^BG, 3β-TCP1/7FastOs^®^BG or 3β-TCP2/7FastOs^®^BG (Figure 1, Table 2). For comparison, the results of autologous bone and BG obtained by us and referred at reference 22 were used.

### 2.5. Ex Vivo Studies

The animals were sacrificed 9 weeks after surgery, and calvaria bone samples were excised. After the sacrifice, an incision was performed with a scalpel #12 in the same location of the previous surgery (day 0) to expose the calvaria bone. After debridement of surrounding soft tissue with a diamond circular cutting blade mounted on a micromotor, 4 perpendicular cuts were made at a sufficient distance from the defects previously created, so that the two defects stay incorporated in the same anatomical piece. A diagonal cut in the distal corner of the left side of the specimen was also made, to enable guiding the sample and identify which side corresponding to each graft substitute used.

### 2.6. Direct Digital Radiography

The excised samples were radiographed using a portable X-ray machine, Port-X II (GENORAY Co. Ltd., 434-6 Sangdaewon-dong, Korea) with an exposure time of 0.04 s under a voltage of 60 kV and a current intensity of 2 mA. A sensor of cesium iodide crystals Gendex (VixWin Pro, version 1.5, 1910 North Penn Road Hatfield, PA 19440 USA) allows the acquisition of the images, which were visualized in a computer monitor and quantitatively analyzed with ImageJ software (NIH, Bethesda, Bethesda, MD 20814, USA). A region of interest (ROI) chosen in each sample was used to obtain the average density values from each bone defect. As a positive control in this evaluation was also drawn an ROI at a non-manipulated bone site.

### 2.7. Histological and Histomorphometric Analysis

The haematoxylin and eosin (H&E) staining (histological analysis) was performed to assess inflammatory infiltrate, fibroblastic proliferation and bone formation. In its turn, histomorphometric analysis was used to evaluate and the % of newly formed bone. Thus, the excised samples were fixed in 10% formalin during 48 h, and decalcified in Osteomoll (EMD Millipore Corporation, Burlington Massachusetts 01803, USA—rapid decalcifier) for about 3 weeks. Afterwards, the samples were embedded in paraffin, sectioned into 5 mm thick sections, and then stained with (H&E) or Von Kossa (VK). The H&E reveal basophilic structures (nucleus) and acidic structures (cytoplasm), respectively. In turn, the VK staining reveals calcium deposits or salts, being therefore useful to confirm the occurrence of mineralization. After staining, the sections that included the defects were subjected to observation under a light microscope (Nikon Eclipse 80i, Postbus 769211070 KE Amsterdam, the Netherlands) and images were acquired using NIS-Elements software (Nikon Instruments Europe BV Postbus 769211070 KE, Amsterdam, the Netherlands). All imaging assessment was performed by a single researcher who was blinded to the sample groups. For histomorphometric measurements Image J (NIH, Bethesda, Bethesda, MD 20814, USA) was used to determine the percentage of newly formed bone (NFB), which was calculated as follows:NFB% = ((New bone area)/TDA) × 100(1)
where TDA corresponds to the dimensions of total defect area.

### 2.8. Statistical Analysis

Statistical analysis was performed using IBM SPSS software v.23.0 (IBM Corporation, Armonk, NY, USA). Normal distribution was assessed by Shapiro-Wilk test and variance of quantitative variables was ascertained by Levene test. Statistical differences were then determined by ANOVA with post-hoc comparison using Games-Howel test, where for *p* < 0.05 the differences were considered statistically significant.

## 3. Results and Discussion

### 3.1. Characterization of the Starting Powders

The XRD patterns of two tricalcium phosphate powders heat treated at 800 °C, one without dopants (TCP0) and the other containing the highest tested concentrations of doping elements (TCP2) are displayed in Figure 2a. The results for TCP1 are not shown as they were very similar to those obtained for TCP2. It can be seen that both TCP0 and TCP2 samples apparently consist of a single crystalline phase corresponding to β-TCP, as all XDR peaks show good coincidence with the diffraction lines of the standard ICDD card number # 04-006-9376 for β-TCP. The XRD pattern of the FastOs^®^BG powder frit is also included, demonstrating its complete amorphous nature.

Other phases such as calcium pyrophosphate (CPP) or HA, when present in amounts below the detection limit, cannot be ruled out by XRD, and their identification requires the use of complementary techniques such as FT-IR. Accordingly, the FT-IR spectra of the TCP powders heat treated at 800 °C are displayed in Figure 2b. The results of FT-IR analysis provide further useful information: a band at 727 cm^−1^ appearing in the spectra of both samples is characteristic of the P-O-P bonds, and typical of the P_2_O_7_^4−^ groups. This band indicates the presence of CPP as minor phase. The presence of bands at 549 cm^−1^, 605 cm^−1^, 941 cm^−1^, 968 cm^−1^, and a broad band within the 1020 and 1120 cm^−1^ [61,62] characteristic of vibrational modes of PO_4_^3−^, confirm the formation of the β-TCP phase as the predominant crystalline phase. The peak 1600 cm^−1^ is due to adsorbed water and the one at 1384 cm^−1^ is assigned to carbonate group (CO_3_^2−^) [62,63]. Therefore, the samples consist of β-TCP as the main phase and of CPP as minor secondary phase.

The particle size distribution (PSD) curves of the starting powders are displayed in Figure 3. These curves show both TCP0 and TCP2 powders consist of bimodal PSDs. The fine particle populations are centered at around 0.4 μm and are likely constituted by individual particles. The coarser populations are centered at around 3 μm and likely represent particle agglomerates incompletely destroyed upon milling. The FastOs^®^BG powder exhibits a considerably extended PSD with only a single main population. The mean particle sizes of TCP0, TCP2, and FastOs^®^BG powder were 1.5 μm, 1.6 μm, and 7.2 μm, respectively.

### 3.2. Postoperative Animal Care

After surgery, there were no identified signs of disease, such as: dehydration, apathy, indifference, prostration, dyspnea, moving in circles or head tipping/tilting, as well as signs of abnormal physiological conditions, in diuresis, faeces volume, body mass and decreased food and/or water intake.

### 3.3. Digital Radiographic Evaluation

The radiographic bone density was analyzed by the Image J software. A region of interest (ROI) in each specimen was chosen and the average value of radiographic bone density from each bone defect was obtained (Figure 4). The data of the radiographic bone density was expressed in arbitrary grayscale media and the results gathered for the samples are plotted in Figure 5.

According to Figure 5, statistical significant differences in the radiographic bone density values can be observed between non-filled defect (Empty group) and defects filled with bone (*p* < 0.05), BG (*p* < 0.05), 3TCP0/7BG (*p* < 0.05), 3TCP1/7BG (*p* < 0.001), 3TCP2/7BG (*p* < 0.001), confirming the regenerative potential of the biomaterials tested. It was also observed a statistical significant difference between empty group and positive control, which confirms the almost null bone regeneration capacity of the unfilled defect during the time studied. There are great similarities in the values obtained for defects filled with autologous bone and with the three investigated synthetic composite bone grafts under study. Although there were no statistically significant differences, there was a tendency for an increase in bone density with increasing of dopant contents. This tendency is likely attributed to the opacifying effects exerted by the doping elements with higher atomic numbers in comparison to Ca = 20 (Sr = 38; Zn = 30, Mn = 25). The absence of doping elements on the case of BG sample is also consistent with its slightly inferior radiographic bone density. However, the measured standard deviation values are relatively high, which difficult to obtain precise conclusions about the effects of the doping elements. The reasons for this high variability might be attributed to: (i) the shallow defects drilled in calvaria bone due to its small thickness; (ii) any possible lateral dislocation of the implanted materials as a result of placing the animals in the lateral decubitus position after surgery in order to facilitate breathing. However, a more detailed analysis about the relative importance of each factor is impossible.

### 3.4. Histological Analysis

Figure 6 shows histological representative images (H&E staining) of the study groups 9 weeks after the treatment and excision. The images of empty bone defects reveal a slight formation of regenerated cancellous bone in the edge of the defect (Figure 6a). In turn, in the BG group (Figure 6b) small islands of osteoid and bone tissue within a capsule are clearly noticed.

The formation of new osteoid and mature bone tissue with many osteoblasts and osteoclasts can be observed. On the other hand, in all experimental groups filled with composites, the formation of new bone tissue with vessels with a pattern of osteogenesis in the defect, containing many of osteoblasts cells could be observed (Figure 6c–e). It is worth noting that, in comparison to defects filled with 3TCP2/7BG (Figure 6e), when the bone defect was filled with 3TCP1/7BG, a larger amount of biomaterial remained non-reabsorbed (white spots) (Figure 6d). The complete absence of inflammation at 9 weeks after the treatment was a common feature to all studied groups.

### 3.5. Histomorphometric Analysis

Through the Von Kossa (VK) staining, the ability of the materials under study for stimulating osteoblastic differentiation was evaluated. In this assay, darker deposits correspond to greater amounts of deposited calcium which, in turn, is an indication of the extent of extracellular mineralization promoted by osteogenesis. Using image processing programs, this qualitative analysis can later give rise to quantitative data, allowing calculating the percentage of newly formed bone. Thus, Figure 6 and Figure 7 represent, respectively, the qualitative and quantitative analysis of newly formed bone in samples excised 9 weeks after the mentioned treatments.

The percentages of newly formed bone in the created defects are displayed in Figure 8, which clearly shows the great advantage of filling bone defects with any of the investigated bone graft materials in comparison to the non-filled defects. As a matter of fact, only a very small amount of newly formed bone was measured in the empty defects.

Statistically significant differences were observed between the empty defect (*p* < 0.001) and all other experimental ones, with the exception of the defect filled with 3TCP0/7BG, where no statistically significant differences were observed. On the other hand, comparing the *in vivo* performance of the different composite bone graft materials, although there are no statistically significant differences, a tendency for an increase in the percentage of new bone formed as the percentage of dopants increases is observed.

Therefore, it can be concluded that the added biologically active inorganic ions (Sr^2+^, Zn^2+^ and Mn^2+^) stimulate the interactions between the composite granules and surrounding cells/tissues, thus accelerating the processes of new bone formation and growth, and of healing the bone defects. Comparing the results obtained for the two composites 3TCP1/7BG and 3TCP2/7BG, the osteogenic potential was slightly higher for the latest one, which can be attributed to its higher concentration in Sr^2+^ and Zn^2+^ [28,29]. In accordance, other studies have demonstrated that the replacement of Ca^2+^ by Sr^2+^ in the crystalline structure of calcium phosphates enhances their osteointegration [28,29], while reducing the reabsorption by osteoclasts, preventing osteoporosis. Other Sr-derived benefits include enhanced bone formation and mechanical resistance, contributing to reducing the risk of fractures [21,33,34]. Regarding zinc, its stimulating roles in bone metabolism, cell adhesion [39], bone formation, regeneration and mineralization are well-known [29,32,33,34,35]. Its inhibitory bacterial growth effect at the surgical site is another important feature for avoiding infections and accelerating wound healing [37,38,39]. On the other hand, the molar concentration of Mn was maintained equal in both metal-doped bone graft materials because of the limited useful ranges of Mn incorporation in terms of *in vitro* biological benefits that can be extracted this doping agent [46].

## 4. Conclusions

The aim of this study was to understand the influence of doping the β-TCP component in the synthetic bone graft composites on the *in vivo* performance in comparison to empty defects, and non-manipulated bone, bone morsels, and FastOs^®^BG used as controls. The results presented and discussed clearly show that all the synthetic biomaterials tested were effective in inducing the bone regeneration. However, the biological response of composite bone graft materials tended to increase with increasing doses of the doping elements, showing that the level of doping is a relevant factor in determining bone density and amount of new bone formed. Accordingly, the best performing composite material [3TCP2/7BG] was as effective as bone morsels and FastOs^®^BG used as control materials. Moreover, after 9 weeks post-implantation, the bone defects regenerated with both 3TCP2/7BG and FastOs^®^BG can hardly be distinguished from the non-manipulated bone. These results are very encouraging towards further testing the most performing materials in future clinical trials. On the basis of the results presented in this study, it appears that doping the β-TCP component in the synthetic bone graft composites may be effective in inducing bone regeneration. This finding confirms that doping the bone graft materials with suitable doses of therapeutic ions is a good strategy towards transposing them to biomedical applications in humans.

## Figures and Tables

**Figure 1 materials-12-00004-f001:**
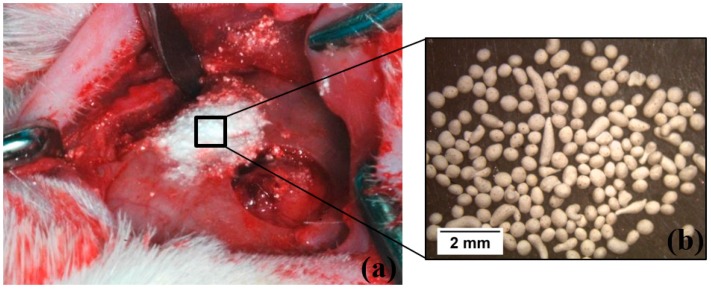
(**a**) Surgical image of the defects created. One defect was empty (negative control group); the other was filled with composite granules shown in (**b**), (experimental group).

**Figure 2 materials-12-00004-f002:**
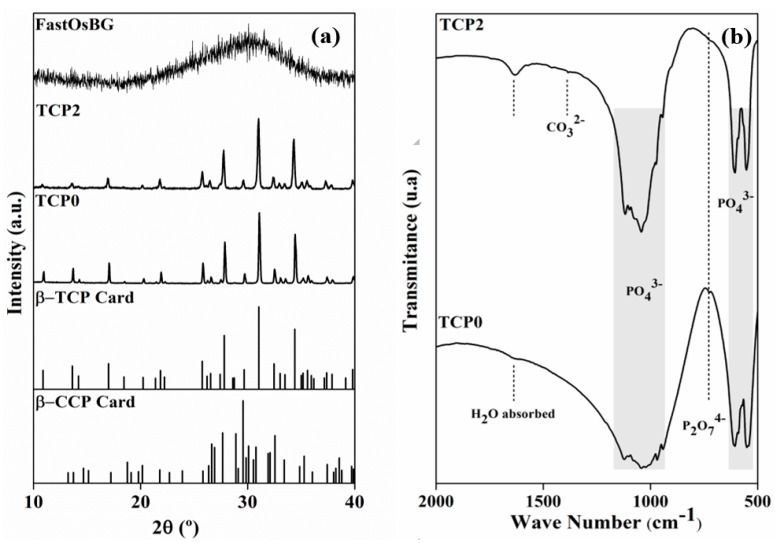
Structural features of the starting powders: (**a**) XRD patterns; (**b**) FT-IR spectra.

**Figure 3 materials-12-00004-f003:**
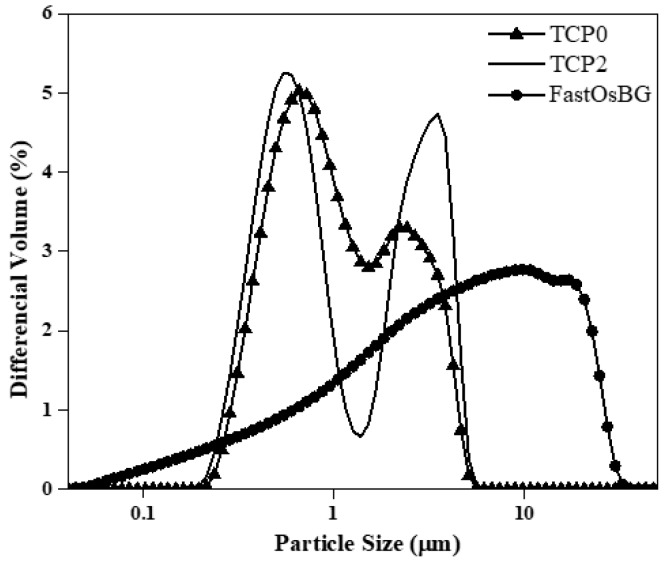
Particle size distributions of the starting TCP0, TCP2 and FastOS^®^BG powders after ball milling.

**Figure 4 materials-12-00004-f004:**
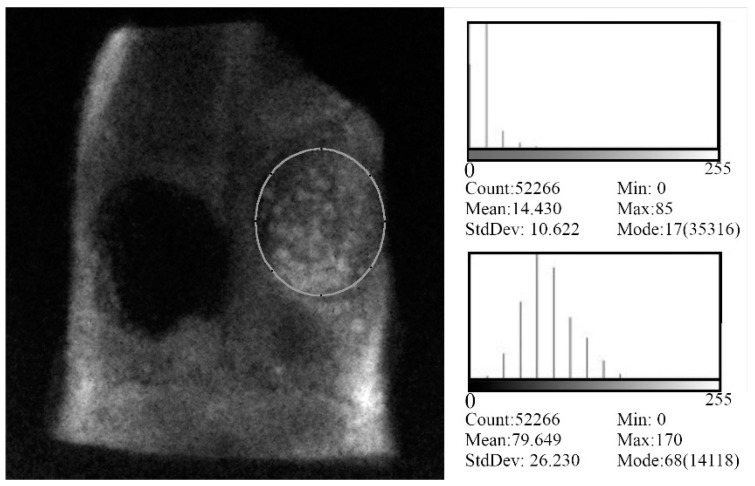
Evaluation of the radiographic bone density and Histogram, 9 weeks after treatment and after excision of bone defects.

**Figure 5 materials-12-00004-f005:**
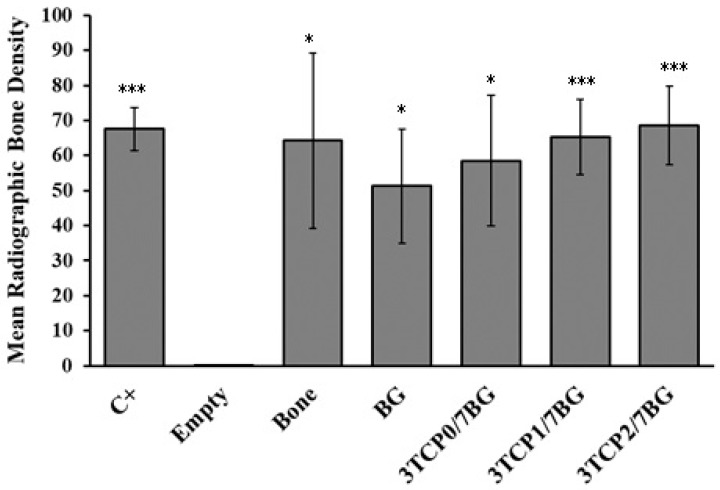
Radiographic bone density of excised bone defects and non-manipulated bone (positive control, C+) evaluated after 9 weeks post-implantation. The figure represents the mean and standard error of at least 5 independent experiments. Statistically significant differences were obtained between the empty defect and the other groups. These differences are identified with the use of * that represents *p* < 0.05 and *** that represents *p* < 0.001.

**Figure 6 materials-12-00004-f006:**
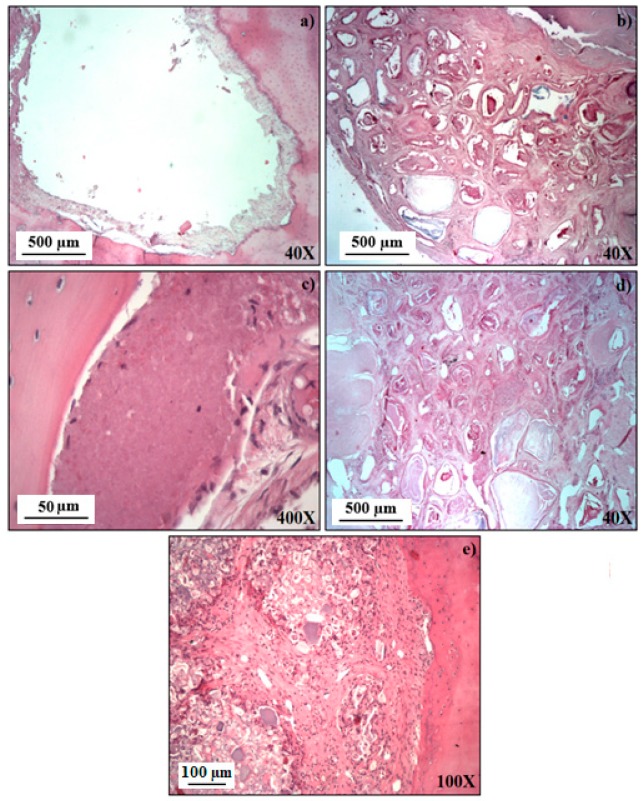
Histological images (H&E staining) of bone defects 9 weeks after treatment: (**a**) empty defect; and defects filled with: (**b**) BG; (**c**) 3TCP0/7BG; (**d**) 3TCP1/7BG; (**e**) 3TCP2/7BG. The Figure shows representative images of at least 5 independent experiments.

**Figure 7 materials-12-00004-f007:**
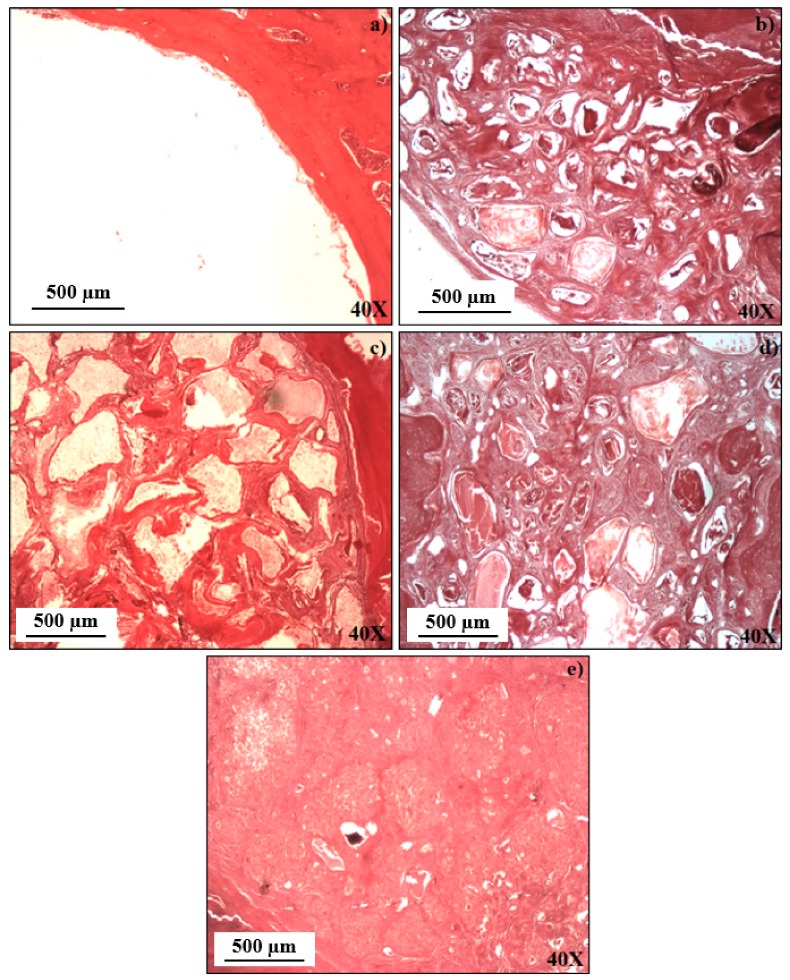
Histological images (VK staining) of bone defects 9 weeks after treatment and after excision: (**a**) empty defect; and defects filled with: (**b**) BG; (**c**) 3TCP0/7BG; (**d**) 3TCP1/7BG; (**e**) 3TCP2/7BG. The Figure shows representative images of at least 5 independent experiments.

**Figure 8 materials-12-00004-f008:**
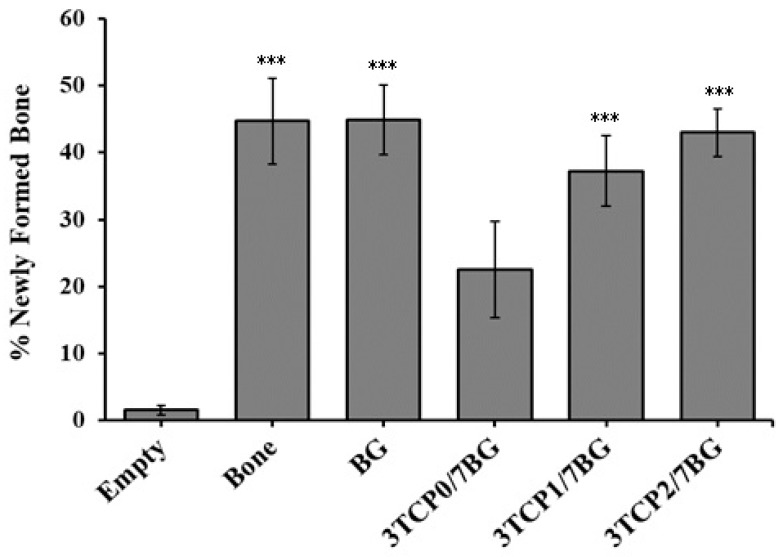
Percentages of newly formed bone in bone defects 9 weeks after treatment and after excision. The Figure represents the mean ± standard deviation of at least 5 independent experiments. Significant differences relative to empty group are identified with the use of *** that represents *p* < 0.001.

**Table 1 materials-12-00004-t001:** Features of the alkali-free FastOs^®^BG.

Features	References
Moderate degradation rate and fast bio mineralization *in vitro*, with HA formation detected after 1 h of immersion in simulated body fluid (SBF)	[23,56,57]
Ability to reduce oxidative stress	[56,57]
Excellent bone bonding ability *in vivo*	[24]
Osteogenic ability	[26]
Excellent sintering behavior	[25,58]
Strong mechanical properties	[25,58]
Easiness of 3D scaffolds fabrication	[59]

**Table 2 materials-12-00004-t002:** Detailed description of the samples, and their respective codes.

Description of the Samples	Codes of the Samples
Component materials	
β-TCP non-doped	TCP0
β-TCP doped with 5Sr, 1Zn, 0.5Mn (mol%)	TCP1
β-TCP doped with 10Sr, 2Zn, 0.5Mn (mol%)	TCP2
FastOs^®^BG	BG
Composites (volume ratio)	
3β-TCP0/7FastOs^®^BG	3TCP0/7BG
3β-TCP1/7FastOs^®^BG	3TCP1/7BG
3β-TCP2/7FastOs^®^BG	3TCP2/7BG
Control groups used in the *in vivo* experiments	
Non-manipulated bone (Positive control)	C+
Empty defect (Negative control)	Empty
Bone-filled defect	Bone
FastOs^®^BG	BG

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
