# Peer review of "Doping β-TCP as a Strategy for Enhancing the Regenerative Potential of Composite β-TCP—Alkali-Free Bioactive Glass Bone Grafts. Experimental Study in Rats"

_materials, 2018, doi:10.3390/ma12010004_

Round 1

Reviewer 1 Report

Dear Authors

This manuscript is well written but needs some more editing for the betterment of scientific soundness. 

Line 35-48: need addition of the following paper, extract information from them and add on the text. 

a) https://www.sciencedirect.com/science/article/pii/S1742706111003497

b) https://www.mdpi.com/1996-1944/8/11/5430

c) https://www.mdpi.com/1996-1944/8/9/5273

Check references carefully. 

Result part can be improved. 

Heading 3.4 and 3.5 can be improved more. 

Author Response

Letter of response to Reviewer #1

Dear Authors

This manuscript is well written but needs some more editing for the betterment of scientific soundness.

Answer: Thanks for the positive appraisal and suggestions. The manuscript was carefully revised with some more editing for enhancing its scientific soundness.

Line 35-48: need addition of the following paper, extract information from them and add on the text.

a) https://www.sciencedirect.com/science/article/pii/S1742706111003497

b) https://www.mdpi.com/1996-1944/8/11/5430

c) https://www.mdpi.com/1996-1944/8/9/5273

Answer: Thanks for the suggestions. Some new research and review papers related with the relevant properties of calcium phosphates have been referred to in the revised manuscript and included in the reference list.

Check references carefully.

Answer: The references have been carefully checked.

Result part can be improved.

Answer: The manuscript was carefully revised and two new Figures were added (Fig. 1 and Fig. 3).  The actual Fig. 2 (previous Fig. 1) was completed with the XRD pattern of bioactive glass to prove its amorphous nature. The results of statistical analysis were also included and discussed.

Heading 3.4 and 3.5 can be improved more.

Answer: The headings 3.4 and 3.5 were improved.

Reviewer 2 Report

Manuscript Review Comments

Title: “Doping β-TCP as a strategy for enhancing the regenerative potential of composite β-TCP–alkali-free bioactive glass bone grafts” (materials-398243)

The authors present a study on the effect of different Beta TCP preparations doped with metal ions in order to get improved results in bone tissue regeneration. However, I find several shortcomings in the manuscript including article that should be addressed.

Comment to the authors

General comments

-          This reviewer found that the presented manuscript has a significative similarity to another manuscript on a different topic previously published by the same authors (Ferreira MM, Brito AF, Marques CF, Freitas LF, Carrilho E, Abrantes AM, et al. Can the regenerative potential of an alkali-free bioactive glass composition be enhanced when mixed with resorbable β-TCP? Ceramics International. 2018;44(5):5025-31.). This reviewer thinks that this fact does not involve the replication of results (although there is some overlap in results section) but the replication of some other parts of the text, fundamentally material and methods, and hence should be rewritten for a structure more compliant and avoiding international potential plagiarism problems.

-          The manuscript needs for a revision of English writing style and grammar, maybe by a native speaker. Typewriting errors, too long sentences, hard to understand, and incorrect enumerations have been observed.

Specific comments:

Title page and abstract:

-          Please avoid the use of acronyms in the title of the article.

-          Please indicate the units in the number between parentheses (line 24).

-          Avoid the use of commercial names in the abstract.

-          The abstract does not report any results or conclusions from the study.

-          Please ensure that the keywords are MeSH terms. Use the appropriate terms as specified in MeSH for better indexation.

Introduction

-          Lines 50-51: Sentence too long. Please split it.

-          Lines 53-55: Sentence too long without connectors.

-          Line 71: “These limitations can be overcome…” Please rephrase. It should be “overcomed”.

-          I find that the introduction gives a great focus on BGs, especially on FastOs, but the objectives do not include it. Also, FastOs BG is also a study group but not considered in objectives. Authors should considerate in reformulating the objectives of the study or making the introduction more oriented to β-TCP and doping elements.

Materials and Methods

-          Is FastOs a commercially available product? Since the authors apply the ® symbol, but later they describe the procedure to make the BG prowder.

-          Please state correctly the commercial reference of the XRD device

-          Please comment on the study design and follow an appropriate guideline for the manuscript preparation and specify this in the methods section. In this case, ARRIVE guidelines are the appropriate for animal studies.

-          The group distribution and study design on rat seems confusing. I recommend a figure presenting a summary of the study with the groups and the follow-up times for more clarity to the reader.

Results and discussion

-          Line 242: Image J does not use a color scale. It is a gray scale.

-          What are the units of radiographic bone density in the Y axis of figure 2? It is %? Since the density was evaluated through the gray scale, I do not understand the figure.

-          Authors do not present any data results from the statistical tests, only state which were significant and which were not.

-          Maybe the author could consider also measuring the Von Kossa results using also Image J analysis and presenting the numeric results from the tests in a table.

Author Response

Letter of response to Reviewer #2

The authors present a study on the effect of different Beta TCP preparations doped with metal ions in order to get improved results in bone tissue regeneration. However, I find several shortcomings in the manuscript including article that should be addressed.

Comment to the authors

General comments

-           This reviewer found that the presented manuscript has a significative similarity to another manuscript on a different topic previously published by the same authors (Ferreira MM, Brito AF, Marques CF, Freitas LF, Carrilho E, Abrantes AM, et al. Can the regenerative potential of an alkali-free bioactive glass composition be enhanced when mixed with resorbable β-TCP? Ceramics International. 2018;44(5):5025-31.). This reviewer thinks that this fact does not involve the replication of results (although there is some overlap in results section) but the replication of some other parts of the text, fundamentally material and methods, and hence should be rewritten for a structure more compliant and avoiding international potential plagiarism problems.

Answer: Thanks for the remarks. The manuscript was carefully revised to properly cope with reviewer’s recommendations with special emphasis on material and methods.

-           The manuscript needs for a revision of English writing style and grammar, maybe by a native speaker. Typewriting errors, too long sentences, hard to understand, and incorrect enumerations have been observed.

Answer: The manuscript was carefully revised and the English writing style and grammar were improved.

Specific comments:

Title page and abstract:

-           Please avoid the use of acronyms in the title of the article.

Answer: The authors understand the reviewer’s concern. But please note that: (i) “β-TCP”, the only acronyms appearing in the title of the article, is very well-known by the scientific community in the field; (ii) it often appears in the title of several articles dealing with calcium phosphates; (iii) it appears twice in the title of this manuscript and its replacement by the full name would exaggeratedly increase the length of the title; (iv) the original title was changed and increased to cope with a comment received from reviewer 3. Therefore, we request to the reviewer to accept the revised version of the title: “Doping β-TCP as a strategy for enhancing the regenerative potential of composite β-TCP–alkali-free bioactive glass bone grafts. Experimental study in rats”.

-           Please indicate the units in the number between parentheses (line 24).

Answer: The units for the numbers between parentheses are molar percentages as already indicated before the parentheses.

-           Avoid the use of commercial names in the abstract.

Answer: Done.

-           The abstract does not report any results or conclusions from the study.

Answer: The abstract was improved to properly cope with this comment.

-           Please ensure that the keywords are MeSH terms. Use the appropriate terms as specified in MeSH for better indexation.

Answer: All the revised key words are MeSH terms, except the “Composite” term in the first one.

Introduction

-           Lines 50-51: Sentence too long. Please split it.

Answer: The sentence was shortened and splinted.

-           Lines 53-55: Sentence too long without connectors.

Answer: The sentence was changed and clarified.

-           Line 71: “These limitations can be overcome…” Please rephrase. It should be “overcomed”.

Answer: The Reviewer is not right here.

-           I find that the introduction gives a great focus on BGs, especially on FastOs, but the objectives do not include it. Also, FastOs BG is also a study group but not considered in objectives. Authors should considerate in reformulating the objectives of the study or making the introduction more oriented to β-TCP and doping elements.

Answer: The introduction was changed and the objectives of the study were better reformulated to cope with this comment.

Materials and Methods

-           Is FastOs a commercially available product? Since the authors apply the ® symbol, but later they describe the procedure to make the BG prowder.

Answer: FastOs®BG is a trade mark. This bioactive glass is under the evaluation process towards getting the CE mark.

-           Please state correctly the commercial reference of the XRD device

Answer: Done.

-           Please comment on the study design and follow an appropriate guideline for the manuscript preparation and specify this in the methods section. In this case, ARRIVE guidelines are the appropriate for animal studies.

Answer: Done

-           The group distribution and study design on rat seems confusing. I recommend a figure presenting a summary of the study with the groups and the follow-up times for more clarity to the reader.

Answer: Done

Results and discussion

-           Line 242: Image J does not use a color scale. It is a gray scale.

Answer: Done

-           What are the units of radiographic bone density in the Y axis of figure 2? It is %? Since the density was evaluated through the gray scale, I do not understand the figure.

Answer: The radiographic bone density in the Y axis of Figure 2 is in %.

-           Authors do not present any data results from the statistical tests, only state which were significant and which were not.

Answer: The data relative to the statistical significance were included.

-           Maybe the author could consider also measuring the Von Kossa results using also Image J analysis and presenting the numeric results from the tests in a table.

Answer: Figure 6 represents the qualitative data obtained from the Von Kossa test. The results of the quantitative analysis of newly formed bone in samples excised 9 weeks after the mentioned treatments are presented in Figure 7. The results of statistical analysis were also included.

Reviewer 3 Report

 Doping β-TCP as a strategy for enhancing the 

regenerative potential of composite β-TCP–alkali-free bioactive glass bone grafts

 Dear authors

Title must be changed into 

 Doping β-TCP as a strategy for enhancing the regenerative potential of composite β-TCP–alkali-free  bioactive glass bone grafts. Experimental study in rats

Materials and Methods section

Can you explain the size of and amount of particles obtained after your dopping procedures?

Please include SEM pimages of material used.

Please include surgical images of your material used in rats.

Please explain why you used ten rats? Why didn’t used 21 rats , so that number after the horles made in rats calvaria make statistical differences? Please explain 

Also I need to know why you didn´t make a critical size defects of 5 mm or more , because the small holes heals quickly. Please explain. Please include some surgical images of your procedure.

The radiographic bone density was analyzed by the Image J software that uses a color scale 242 with 256 levels, including zero for the black color, 254 different gray and the 255 is white.Pleases include in the paper  some descriptive images.

Why the rats were sacrified at 9 weeks and you didn’t sacrified at different time procedures? Please Explain

Results section

Must be improved

Discussion section

You must discuss your results with those papers

Comparison of Two Xenograft Materials Used in Sinus Lift Procedures: Material Characterization and In Vivo Behavior.

Ramírez Fernández MP, Mazón P, Gehrke SA, Calvo-Guirado JL, De Aza PN.

Materials (Basel). 2017 Jun 7;10(6). pii: E623. doi: 10.3390/ma10060623.

Conclusions

Please explain in detail the clinical relevance after your research to be traspole to a human 

Author Response

Letter of response to Reviewer #3

Dear authors

Title must be changed into

Doping β-TCP as a strategy for enhancing the regenerative potential of composite β-TCP–alkali-free bioactive glass bone grafts. Experimental study in rats

Answer: The title was changed as suggested.

Materials and Methods section

Can you explain the size of and amount of particles obtained after your dopping procedures?

Answer: The particle size distributions of the different starting powders are now displayed in Fig. 3.

Please include SEM pimages of material used.

Answer: An optical image representative of the granular bone graft materials is now shown in Fig. 1(b).

Please include surgical images of your material used in rats.

Answer: Fig. 1(a) presents a surgical image of the defects created. The empty defect (negative control group), and the other one filled with the composite granules shown in (b), (experimental group).

Please explain why you used ten rats? Why didn’t used 21 rats, so that number after the horles made in rats calvaria make statistical differences? Please explain

Answer: The use of non-critical defects (4 mm) allows performing two bone defects in the calvaria of each animal. Thus, with 10 animals 20 bone defects are achieved, which allows us to decrease the number of animals used, against one of the principles of 3R (replacement, refinement and reduction) policy.

Also I need to know why you didn´t make a critical size defects of 5 mm or more, because the small holes heals quickly. Please explain. Please include some surgical images of your procedure.

Answer: Critical size defects were not made because in the calvarial model of wistar rats, the mortality rate is high when such defects are made. In addition, the size of the defect chosen allows two defects to be made in one animal, thus reducing the number of animals to be included in the test, this procedure is in line with the 3R (replacement, refinement and reduction) policy. The unfilled bone defect was then used as a negative control. This option proved to be valid since at the time of animal sacrifice the percentage of new bone in the empty defects was practically zero.

The radiographic bone density was analyzed by the Image J software that uses a color scale 242 with 256 levels, including zero for the black color, 254 different gray and the 255 is white. Pleases include in the paper some descriptive images.

Answer: Thanks for remark. The required changes were made.

Why the rats were sacrified at 9 weeks and you didn’t sacrificed at different time procedures? Please Explain

Answer: This time of sacrifice was chosen taking into account the experience of the research group with this animal model. In previous studies aiming at optimizing the implantation time, this time point was found to be the ideal one to enable a good balance between the non-visualization of new bone formation in the empty defect and the formation of new bone when the defects are filled with autologous bone (gold standard).

Results section

Must be improved

Answer: The manuscript was carefully revised giving particular attention to the Results section.

Discussion section

You must discuss your results with those papers

Comparison of Two Xenograft Materials Used in Sinus Lift Procedures: Material Characterization and In Vivo Behavior.

Ramírez Fernández MP, Mazón P, Gehrke SA, Calvo-Guirado JL, De Aza PN.

Materials (Basel). 2017 Jun 7;10(6). pii: E623. doi: 10.3390/ma10060623.

Answer: The discussion was improved and our results were compared with other relevant available literature reports. Xenograft materials are also interesting, but are out of the scope of the present work.

Conclusions

Please explain in detail the clinical relevance after your research to be traspole to a human

Answer: The conclusions were improved to cope with this comment.

Round 2

Reviewer 1 Report

Dear Authors Just give one final review for English language and style are fine and also check minor spell. You all done a constructive change on revision. Well appreciated.

Reviewer 2 Report

Authors have addressed most of the queries established. I consider this paper now valid for publication.

Reviewer 3 Report

THE PAPER IS CORRECT